# Integration of Inkjet Printed Graphene as a Hole Transport Layer in Organic Solar Cells

**DOI:** 10.3390/mi14101858

**Published:** 2023-09-28

**Authors:** Julia Kastner, Flavia Tomarchio, Nicolas Decorde, Matthias Kehrer, Günter Hesser, Anita Fuchsbauer

**Affiliations:** 1Functional Surfaces and Nanostructures, Profactor GmbH, 4407 Steyr-Gleink, Austria; 2Cambridge Graphene Centre, University of Cambridge, Cambridge CB3 0FA, UK; 3Center of Surface- and Nanoanalytics, Johannes Kepler University, 4040 Linz, Austriaguenter.hesser@jku.at (G.H.)

**Keywords:** graphene, inkjet printing, organic solar cells, hole transport layer

## Abstract

This work demonstrates the green production of a graphene ink for inkjet printing and its use as a hole transport layer (HTL) in an organic solar cell. Graphene as an HTL improves the selective hole extraction at the anode and prevents charge recombination at the electronic interface and metal diffusion into the photoactive layer. Graphite was exfoliated in water, concentrated by iterative centrifugation, and characterized by Raman. The concentrated graphene ink was incorporated into inverted organic solar cells by inkjet printing on the active polymer in an ambient atmosphere. Argon plasma was used to enhance wetting of the polymer with the graphene ink during printing. The argon plasma treatment of the active polymer P3HT:PCBM was investigated by XPS, AFM and contact angle measurements. Efficiency and lifetime studies undertaken show that the device with graphene as HTL is fully functional and has good potential for an inkjet printable and flexible alternative to PEDOT:PSS.

## 1. Introduction

Organic photovoltaics (OPV) belongs to the area of thin-film technologies of solar cells and had its beginnings in the early 1990s, where mostly fullerene derivates were used for the active layer [1]. Subsequently, with the help of improved technology and materials, power conversion efficiencies up to more than 18–19% have been achieved [2,3,4,5,6,7,8,9,10]. High efficiencies can be achieved with active layers based on narrow bandgap non-fullerene acceptors (such as Y6 [11] and M3 [12]) and blends like PM6:Y6 [3,13], D18:Y6Se [14] or PM6:L8-BO [15]. Moreover, currently, ternary and quaternary organic solar cells are also being investigated [2,6,16,17], where a third or fourth material as an additional acceptor and/or donor is introduced into the photoactive layer for improving photon capture, helping to optimize the distribution of generated excitons and supporting the degree of phase separation [18].

Organic semiconductors can be processed in solution [19,20,21]. Therefore, these cells are compatible with large-scale and flexible roll-to-roll production, since they are fully printable [22,23,24,25]. Inkjet printing is a method for digital printing of conductive or any functional inks [26,27,28,29,30,31]. Therefore, inkjet printing of the different interlayers in OPV devices is also of great interest [32,33,34,35,36,37]. The main function of interlayers in OPVs is to prevent charge carrier recombination at the interface by blocking one type of charge carrier. They establish an ohmic contact and improve charge carrier extraction at the metal/organic interface. Moreover, optimization of the built-in potential and film morphology plays an important role. Therefore, an electron transport layer (ETL) at the cathode interface and a hole transport layer (HTL) at the anode interface are introduced in both conventional (low work function as top electrode) and inverted device geometries (with high work function material as the top electrode) of bulk hetero-junction (BHJ) solar cells (see schemes in Figure 1). 

Depending on the different geometries and photoactive layers used, different materials can prolong the lifetime and stability of the solar cell [38]. 

For example, commonly used ETL materials include ZnO or modified ZnO in inverted organic solar cells (OSCs) [39] and, recently, organic compounds, like N,N′-Bis{3-[3-(Dimethylamino)propylamino]propyl}perylene-3,4,9,10-tetracarboxylic diimide (PDINN), N,N-dimethyl-ammonium N-oxide)propyl perylene diimide (PDINO) and PDINO mixed with polyethylenimine ethoxylated (PDINO:PEIE) in non-fullerene conventional devices [18,40,41].

HTLs used in organic solar cells include poly(3,4-ethylenedioxythiophene) polystyrene sulfonate (PEDOT:PSS) [36,42,43,44,45], and, in inverted architectures especially, MoO_3_ and transition metal disulfides or other oxides [18,46,47]. Usually, the oxides are applied by thermal evaporation. Solution processing of the metal oxides is still a challenge and can lead to faster degeneration in inverted architectures [38,46].

Graphene has also been heavily investigated for applications in OPV. Several publications have reported the usage of graphene in organic solar cells, mostly used as graphene oxide or composites, and for solar absorbers [36,48,49,50,51]. The advantage of graphene oxide (GO) is that it is prepared in solution (Hummer’s method to force exfoliation of graphite by oxidation [52,53,54]) and can be easily functionalized to tune the work function from 3.7 to 5.1 eV [55,56,57] with hydroxyl and carboxyl groups [35]. GO is non-toxic; however, it can develop toxic gases during production and shows low conductivity [53,58,59]. Reduced GO (rGO) can overcome the problem of low conductivity; however, it is prepared from GO by either high energy consumption methods [53] or harsh chemical reduction using, for example, the highly toxic hydrazine [60]. 

In the case of an inverted structure (Figure 1b), the stability of the solar cells can be increased radically compared to the conventional geometry (Figure 1a) due to different factors, like better chemical stability of the higher work function metals used as the top contacts [61]. PEDOT:PSS is a commercially available, transparent (>80% transparency), and conductive (sheet resistance ~40 Ω/□) polymer [62] and is still widely used as an HTL. However, due to its sensitivity to moisture and oxygen [61,63], it can limit lifetime in combination with other degradation factors [64]. The hygroscopic nature of PEDOT:PSS promotes absorption of ambient water inside the solar cell and, thus, can lead to a conductivity decrease in PEDOT:PSS [65]. 

In this work, we tested liquid phase exfoliated (LPE) graphene material as an HTL [66], taking advantage of the high electrical conductivity and stability of graphene [67]. Furthermore, liquid phase exfoliation of graphite is a promising method for producing a graphene dispersion that is simultaneously scalable, water-based, and that can be performed at room temperature, offering a high yield [68] compared to GO or rGO. Graphite was exfoliated in water with the acid bile salt sodium deoxycholate (SDC) to enhance exfoliation and to stabilize the graphene nanoparticles in water [69]. An additional advantage of SDC is its non-toxicity for humans and the environment. For solar cell evaluation, we replaced PEDOT:PSS on a fullerene-based active layer for initial tests to determine the potential of the exfoliated graphene flakes as a charge carrier in first lifetime tests. This was not to achieve high efficiencies, but for comparison with another frequently used printable material.

## 2. Materials and Methods

A quantity of 12 g/L of natural graphite flocks was exfoliated by liquid phase exfoliation (LPE) in water with 9 g/L sodium deoxycholate (SDC). Subsequently, the dispersion was ultracentrifuged, and the supernatant collected was the graphene standard ink. Iterative centrifugation was used to concentrate the graphene ink. The concentration of the dispersions was determined by optical absorption spectroscopy (OAS) with absorption coefficient α = 1390 L/g·m at 660 nm [70]. A final concentration of 1 g/L was obtained after iterative centrifugation (all details on preparation and characterization are provided in the Appendix A).

Raman spectra were acquired at 457, 514 and 633 nm using a Renishaw InVia spectrometer (Renishaw GmbH, 72124 Pliezhausen, Germany) equipped with a 100× objective (N.A. = 0.85). Ca. 20 spectra were collected for statistical analysis.

Graphene solar cell devices were prepared by inkjet printing graphene on an ITO/ZnO/poly(3-hexylthiophene mixed with a phenyl-C-61-butyric acid methyl ester (P3HT:PCBM) stack, which was made by doctor-blade-coating under ambient conditions. Laser-structured substrates of ITO were cleaned with a sequence of successive baths in acetone and IPA. The ZnO layer (ZnO suspension from Ibu Tec) was blade-coated and annealed for 5 min at 130 °C in air. The active layer was created by mixing P3HT with PCBM (both from Merck) 1:0.8 weight ratio in o-xylene and tetrahydronaphtalene with a ratio of 9:1. For reference cells, PEDOT:PSS (1:3 Clevios HTL-388:IPA) was doctor-bladed as an HTL with a temperature of 65 °C and annealed at 130 °C for 5 min in an inert atmosphere. Ag (100 nm) was evaporated at a standard rate of 1–5 Å/s.

Before printing, the surface of the substrate was exposed to argon plasma in a “nano” plasma asher from Diener electronic GmbH & Co.KG (Ebhausen, Germany). The plasma-treated P3HT:PCBM surface was characterized by the surface free energy (SFE) and contact angle (CA) using a Krüss DSA 100 goniometer. SFE was calculated from the measured contact angles of water, diiodo-methane, and ethylene glycol. Atomic force microscopy (AFM) measurements using a Bruker Dimension Edge operated in tapping mode were used to acquire the surface topographics. The surface roughness was calculated from 20 × 20 µm^2^ scans with the average value from three different samples taken. XPS measurements were carried out on a Theta Probe XPS system (ThermoFisher, Waltham, MA, USA, GRB) using a monochromatic Al-Kα X-ray Source with a photon energy of 1486.6 eV. For survey scans and high resolution (HR) scans, the hemispherical analyzer was set to a pass energy of 200 eV and 20 eV. The energy step sizes were 1 eV and 0.05 eV, respectively.

The graphene ink was printed in an ambient atmosphere, but in rooms with UV-reduced light conditions, on the pre-fabricated ITO/ZnO/P3HT:PCBM substrates with a Dimatix Printer DMP-2831 from FUJIFILM Dimatix, Inc. (Lebanon, NH, USA) and a drop volume of 10 pl, with a drop space of 65 µm. After printing, the stabilizing surfactant SDC was removed by rinsing with a mixture of H_2_O:ethanol 1:1 solution for few seconds. Transmission electron microscopy (TEM) measurements were performed with a JEOL JEM-2011 (from JEOL GmbH, Freilassing, Germany) equipped with a LaB_6_ filament. Images were taken with a Gatan Slow-Scan CCD Camera Model. For the energy-dispersive X-ray spectroscopy (EDX) measurements, a JEOL JEM-2200FS TEM equipped with an OXFORD Instruments EDX system with an SDD Detector (Oxford X-Max^N^ 80T”, 80 mm^2^, Abingdon, UK) was used.

The current–voltage characteristics of the solar cells were recorded under a calibrated Oriel solar simulator with a xenon lamp (AM1.5G, 1000 W/m^2^, 1 Sun intensity) and measured with a Keithley 2400 source meter in combination with a Keithley 7001 Multiplexer system.

For stability testing, the samples were stored in a dedicated damp heat (DH) chamber, encapsulated under open-circuit conditions. The climate conditions were set to 85 °C and 85% relative humidity (rH). For every measurement, the samples were taken out of the chamber, allowed to cool, and measured in an inert atmosphere with the solar simulator.

## 3. Results and Discussion

### 3.1. Characterization of Graphene Flakes

The Raman spectrum of graphene consists of a set of distinctive peaks; the D, G and 2D peak are the fingerprints for graphene (Figure 2a). In single-layer graphene (SGL), the 2D peak is a single Lorentzian, whereas it splits into several components as the number of layers increases, reflecting the evolution of the electronic band structure [71]. The line shape of the measured 2D bands of the graphene flakes showed a single Lorentzian curve. Their positions Pos(2D) were at ~2698.5 cm^−1^, and the full width at half maximum FWHM(2D) was ~64 cm^−1^. The I(2D)/I(G) (Figure 2b) ranged from 0.4 up to 1.04. These values suggest that the sample was a combination of mono-layers and a few layers of graphene, which were electronically decoupled such that, to a first approximation, they behaved as a collection of mono-layers [72].

The Raman spectra showed significant D and D′ peak intensities, with I(D)/I(G) ranging from 0.7 to 2.3. However, the value of the rate of change of the G peak position with excitation wavelength Disp(G) of ~0.006 cm^−1^/nm was much lower than the value expected for disordered carbon [73]. Therefore, the D peak was attributed to the edges of submicrometer graphene flakes, rather than to the presence of a large quantity of structural defects within the flakes [74]. This observation was supported by the absence of a correlation between I(D)/I(G) as a function of Disp(G). 

### 3.2. Fabrication of Solar Cells with Graphene Ink as a Hole-Transport-Layer

The as-prepared graphene ink was used as the HTL of the organic solar cells with an inverted device configuration of ITO/ZnO/P3HT:PCBM/graphene/Ag (compare Figure 1b). The solar active polymer P3HT:PCBM as the underlying layer for the HTL was very hydrophobic. This generally causes lot of problems—poor adhesion and delamination between the active polymer and the normally used PEDOT:PSS [75]. In order to obtain a continuous film with the water-based graphene ink, the surface energy was modified by plasma etching. Since P3HT:PCBM is very oxygen sensitive [76], activation with argon (Ar) plasma was tested. An annealing step prior to the plasma treatment at 130 °C for 3 min was performed to eliminate the formation and widening of cracks in the P3HT:PCBM layer and to make it more resistant to further processing. Mild (15 W, 6 s) and harsh (240 W, 1 min) argon plasma treatment after annealing significantly increased the surface free energy to 36.7 mN/m and 55.8 mN/m, respectively, ensuring optimal wetting. The surface roughness of the plasma-treated substrates was measured by AFM (see Appendix A). The measured root mean square roughness (*R_rms_*) showed no significant difference for all substrates and was 12.2 ± 1.0 nm, 12.5 ± 3.1 nm and 13.7 ± 1.8 nm for untreated, mild plasma-treated and plasma-treated P3HT:PCBM substrates, respectively, which is in accordance with the literature [77,78]. XPS measurements were performed to further investigate the treated surface (see Appendix A). The measurements revealed oxidation of the P3HT:PCBM surface after Ar plasma treatment: the O 1s peak was significantly increased from 3 atomic% for the untreated reference to 14 atomic% for the mild plasma-treated and 21 atomic% for the 240 W plasma-treated P3HT:PCBM surface. With increasing plasma power and time, more carbonyl and carboxyl species were formed, matching the observation of increase in the polar part of SFE measurements, while the C-S-C bound was heavily reduced. This led to a barrier in the solar cell with no proper charge transport [79,80]. Solar cells with a doctor-bladed PEDOT:PSS layer and an evaporated silver electrode to test the photovoltaic performance of the plasma-treated devices showed a second diode behavior with a very high threshold voltage needed for the harsh plasma-treated substrate, supporting the results of the XPS measurement. The samples treated with mild plasma conditions showed no significant changes in the device performance and were used for further inkjet printing tests of the graphene ink as an HTL on the active polymer.

AFM measurements taken of the inkjet-printed graphene film showed an increase in the *R_rms_* from 12.5 nm of the mild plasma-treated P3HT:PCBM substrate to 23.6 ± 0.8 nm for 10 inkjet printing passes of the graphene ink and of 19.1 ± 1.9 nm for 20 printing passes, respectively. Interestingly, the roughness of the graphene film could be reduced slightly by printing more layers. This was attributed to a more homogenous distribution of printed graphene flakes on the substrate. The transparency of the printed graphene film on P3HT:PCBM was measured by OAS. The value obtained at 550 nm was ~75–80%.

After the deposition of the Ag top electrode, the morphology of the unencapsulated finished solar cells with graphene film as an HTL was investigated with transmission electron microscopy (TEM). Figure 3 shows a cross-section of an organic solar cell produced within this work. The magnification shows that the graphene flakes were well aligned on the hydrophobic substrate. Moreover, no Ag particle formation was observed. Energy-dispersive X-ray spectroscopy (EDX) measurements (see Appendix A) clearly showed that Ag remained above the graphene layer with no Ag particle diffusion into the P3HT:PCBM material, although the cells were neither encapsulated nor stored in an inert and dry atmosphere for two months.

### 3.3. Evaluation of the Solar Cells and Lifetime Studies

Different cells with printed graphene as an HTL were fabricated with an increasing number of printing passes from 10 up to 20. Control devices with PEDOT:PSS (doctor-bladed) as the HTL were also fabricated under identical conditions. The photovoltaic performances of these devices are summarized in Table 1; Figure 4 shows the typical corresponding *J-V* curves.

As shown by these data, the performance of the solar cells changed with the number of printing passes of the graphene ink, although none of the graphene cells could reach the reference cell with PEDOT:PSS. Measured dark *J-V* curves showed that the samples with graphene suffered from lower parallel resistances than the reference. This points to small pinholes in the graphene layer where the top Ag electrode can have a direct contact with the active layer; however, no direct trend with printing passes could be detected. The best performance was achieved with 15 passes and a maximum efficiency of 2%. The appearance of an s-shape in the *J-V* curves from the graphene samples with 10 and 20 printing passes was notable. 

The devices with 10 printing passes of graphene showed similar *J_SC_* and *V_OC_* values as the device with 15 printing passes; however, the FF was lower. This indicates that a thinner layer of graphene material results in lower conductivity, resulting in a higher series resistance *R*_s_ which lowers the FF [81]. Moreover, the low slope of the *J-V* curve may be due to no good current transport.

The *J_SC_*-value of the device with 20 printing passes of graphene was remarkably high at 10 mA/cm^2^ compared to the reference device. The thicker graphene layer probably promotes an optical spacer effect [82,83], but was not further investigated in this work. A transparent material layer with a certain layer thickness (e.g., ZnO at about 10 nm) inserted between the reflecting metal electrode and the active layer can act as a so-called optical spacer. It helps to enhance the light absorption by changing the position of the local absorption maximum in the photoactive layer, which increases the charge collection. However, in these graphene devices, most probably, the insufficient removal of the SDC surfactants reduced the efficiency by influencing the work function and limiting the current transport, which was also indicated by the lifetime studies described below. 

Ideally, the work function of the HTL material should be ca. −5.0 eV for proper charge transport. Measurements with a Kelvin probe set-up (Single Point Kelvin Probe set-up from KP Technology) showed a work function of the inkjet-printed graphene on ITO of −4.88 ± 0.036 eV. In the literature, tuning of the work function of graphene is described, e.g., via changing the electrical field and using the electric field-effect [84,85], exposing to irradiation [86] or gases [87,88,89], and functionalization [90]. Literature studies also indicate that the electronic properties of graphene are highly sensitive to water (and, therefore, ambient air) [82,91] and the underlying substrate [82,92], but are also dependent on the number of graphene layers [93]. Melios et al. [82] showed that increasing humidity resulted in a substantial increase in work function and hole concentration with saturation at about 40% relative humidity. They also indicated that, in ambient air, in addition to water, further p-dopants are present. Moreover, the work function of graphene is probably substrate-dependent. For example, native SiO_2_ lowers the hole concentration in suspended CVD graphene. 

To test the stability of the graphene solar cells, lifetime studies were conducted. DH conditions of 85 °C and 85% rH without any encapsulation were used to accelerate the degradation and to monitor the decay. Cells with 10 and 15 printing passes of graphene and PEDOT:PSS as an HTL as reference cells were tested in DH for 900 h and measured in between. A total of 16 out of 24 graphene cells and 5 out of a total of 8 PEDOT:PSS reference cells showed typical degeneration and were used for the evaluation. Moreover, a further 8 reference cells were also treated with the same pre-treatment procedure as used for the samples with the graphene (annealing and subsequent Ar plasma) before the application of PEDOT:PSS. Interestingly, these cells showed very poor characteristic *J-V* curves and degraded very fast, whereas the lifetime of the cells with graphene seemed to be unaffected by the pre-treatment itself (compare Figure 5a). This might have been due to poorer wetting behavior of the PEDOT:PSS on the active polymer after plasma treatment. This promotes easier delamination during DH conditions, which results in a fast loss of current flow. Therefore, in the following results the cells with printed graphene were only compared to the untreated reference cell. A drastic increase in the efficiency of the graphene cells was observed after the first 18 h in the DH chamber. This effect will be discussed in the following paragraph. Due to this peak, normalization of the graphs was performed from the measuring point after 18 h (Figure 5b and Figure 6).

Figure 6 shows the comparative decay between the untreated reference solar cells and the graphene devices of the normalized *J_SC_*, *V_OC_*, FF under DH storage. The use of deeply studied materials (ITO/Zn/P3HT:PCBM/PEDOT:PSS/Ag) as a reference was advantageous for comparing the degradation.

Figure 6a demonstrates that the short circuit current decreased similarly for both samples, which can be mainly attributed to the degradation in oxygen of the semiconductor (P3HT:PCBM) or, more likely, of the ETL ZnO [94,95].

Also *V_OC_* (Figure 6b) decreased with the same trend as the reference devices. The difference in work functions of the top and bottom electrodes created an internal or so-called built-in electric field in the OPV device, which facilitated the transfer of the respective charge carriers to the electrodes and, therefore, prevented recombination losses [96,97]. *V_OC_* describes the voltage at the point of no current flow. Here, the electric field is too low to further separate the excitons at the acceptor-donor interfaces. This is generally dependent on the work function alignment and the difference in the potential of the donor and acceptor materials used in the photoactive layer and the electrodes for highly efficient charge extraction [98,99]. The comparable *V_OC_* drop in both the graphene and reference devices was mostly due to a change in morphology and growth in deeper carrier traps caused by oxidation of the photoactive layer [38,95,100]. These traps disrupt the distribution of the internal field and can enhance the charge carrier recombination. However, the initial change in the graphene devices after the first 18 h was drastic in the *V_OC_* data. Figure 7a,b show the evolution of representative *J*-*V* curves during storage at DH storage. While the reference cells (Figure 7a) showed typical degeneration [101], the graphene cells, at the beginning (0 h), showed a slight sigmoidal or s-shaped curve, which vanished after the first 18 h after storage in DH (compare Figure 7b). Both observations indicate better carrier extraction at the anode. and point to a change in the work function of either or both the HTL and the electrode. Since the work function of silver only changed from −4.3 eV auf −4.5 eV during aging [102], most likely the work function of the graphene layer was changing. As discussed above, the work function of graphene increases in humid atmosphere and, additionally, surfactants, like the remaining SDC from the exfoliation process, cause changes to the work function [103]. However, SDC is water-soluble and can diffuse out of the layer in a humid atmosphere. As confirmation, the sheet resistance measurements of the printed and washed graphene layers on glass and OPV indeed showed improvement in the conductivity of about 8% after storage in a warm humid atmosphere (85 °C, ~90% rh) for 24 h. This resulted in the better work function and lower series resistance, as discussed in the following paragraph.

The FF (Figure 6c) times the product of *V_OC_* and *J_SC_* gives the maximum power delivery by the solar cell. As for the *V_OC_*, the FF increased for the graphene samples in the first 18 h in DH. However, a slight increase in the performance for the reference cells was also observed. This can be attributed to an annealing effect, where enlarged PCBM domains in the mixture lead to improved electron transport properties in the active layer, whereas further annealing leads to morphological deterioration due to too large PCBM domains [104]. However, the FF of the graphene sample, after an initial decrease, stayed almost the same over the complete time period, remaining at 85% of the initial value compared to 58% of the FF value of the cells with PEDOT:PSS. The FF is mostly effected by the series resistance *R*_S_ in the cell set-up [105]. The *R*_S_ can be calculated from measured dark curves [106] or the slope of the illuminated *J-V* curves at *V* = 1.5 where the current is mostly limited by *R*_S_ and the *J-V* curve becomes linear [97]. *R*_s_ consists of a sum of all the series resistances along the solar cell from the transport in the active layer, the resistances in the interlayers, and the final transfer through the electrode contacts [104]. Ideally, the series resistance should be as low as possible to maximize the increase in current during forward bias by improving the charge extraction to the respective electrodes [107]. For example, this value is very high in devices using graphene oxide (GO) as a hole transport layer, but is decreased with graphene/GO composites due to the better vertical conductivity of graphene [105].

Since graphene cannot degrade as a polymer like PEDOT:PSS, *R*_S_ did not increase in the graphene devices (compare Figure 7c). While the *R*_S_ of the graphene device stayed around the initial value (~9.6 Ω cm^2^), the *R*_S_ for the reference sample continuously increased from an initial average value of 2.7 Ω cm^2^ to 62.7 Ω cm^2^ after 645 h in the harsh damp heat conditions. This could be the reason for the drastic failure of the *J-V* curves of the reference samples beginning after 474 h (Figure 7a), while the *J-V* curves of the graphene samples only showed a slight degeneration compared to the initial shape after 18 h in DH.

Another reason for improved stability in the graphene sample could be the above-described non-forming of silver particles, or that the interface and adhesion between the semiconductor and the graphene was better (similar surface energy) than that between the semiconductor and PEDOT:PSS. It has to be taken into account that almost 90% of all the tested cells with graphene were still functioning at the end of the lifetime testing, whereas half of the reference solar cells had completely degenerated.

All these findings show that in liquid, exfoliated graphene is a prospective hole-transport layer, which shows potential to increase stability in OPV cells. However, to achieve reliable results, the SDC has to be decreased in the final printing solution or the washing process has to be improved. A solution could be to replace all, or at least part, of the SDC with additives, which can also act as p-dopants, to slightly increase the work function and the hole concentration of the graphene.

## 4. Conclusions and Outlook

A water-based graphene ink was produced via ultrasonication. By centrifugation, the ink was purified from large graphite particles and subsequently concentrated to 1 g/L. The concentration was confirmed by optical adsorption spectroscopy and the graphene flakes were characterized by Raman.

It was shown that water-based graphene ink can be easily inkjet printed on Ar plasma-treated P3HT:PCBM substrates for usage as an HTL in inverted structure organic solar cells. The procedure described in this work has enabled, for the first time, the production of working solar cells with inkjet-printable graphene (not graphene oxide) as the HTL. The solar cells showed a similar long-term stability as solar cells with commercial doctor-bladed optimized PEDOT:PSS when no encapsulation was used. In particular, the persistence of the series resistance for more than 640 h shows the high potential for improving the stability performance of organic solar cells with graphene. This study indicates that printed graphene flakes produced by exfoliation in green solvents without the use of toxic and critical components can serve as a stable, low cost and flexible alternative, allowing the production of fully inkjet-printable organic solar cells.

After initial tests to assess the potential of the graphene flakes and first lifetime tests, further in-depth tests, such as light-intensity-dependent *J-V* tests, and in-depth work on work function improvement and investigation of graphene flakes (including doping experiments), will follow in the future. Light intensity tests would enable checking the effects of several factors, i.e., if a lower FF is just due to low *R*_s_ (if yes, the FF will increase with lower light intensities) or if other factors play a role (i.e., different kinds of recombination). Additionally, graphene as an HTL will be tested on current ternary semiconductor systems, such as PM6:Y6:PCBM.

## Figures and Tables

**Figure 1 micromachines-14-01858-f001:**
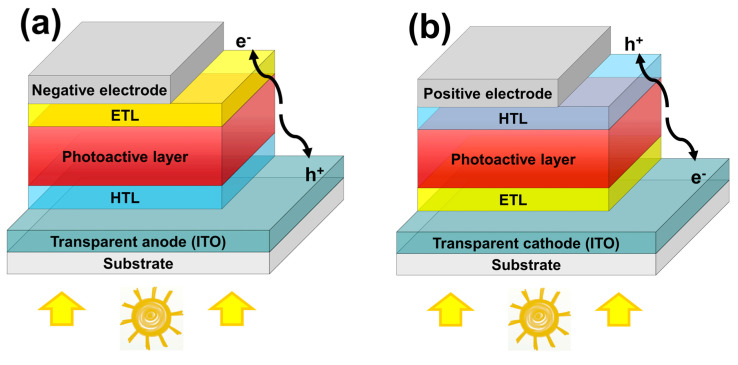
Scheme of organic solar cells with (**a**) conventional geometry and (**b**) inverted geometry.

**Figure 2 micromachines-14-01858-f002:**
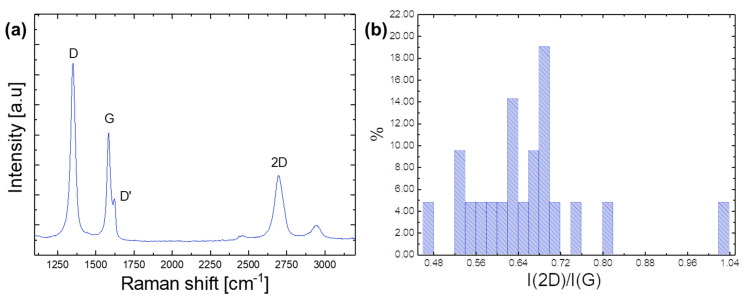
(**a**) Representative Raman spectra of the graphene ink by LPE at 514.5 nm. (**b**) Distribution of I(2D)/I(G).

**Figure 3 micromachines-14-01858-f003:**
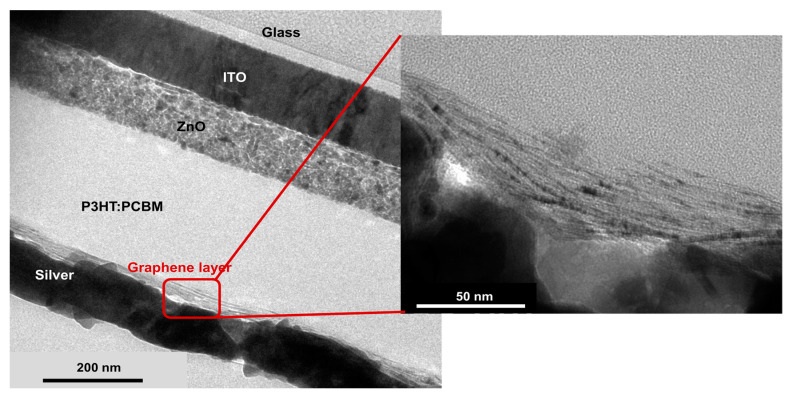
TEM image of cross-section of a solar cell with inverted structure glass/ITO/ZnO/P3HT:PCBM/graphene/silver. The insert shows the clearly visible graphene flakes between the silver and the P3HT:PCBM.

**Figure 4 micromachines-14-01858-f004:**
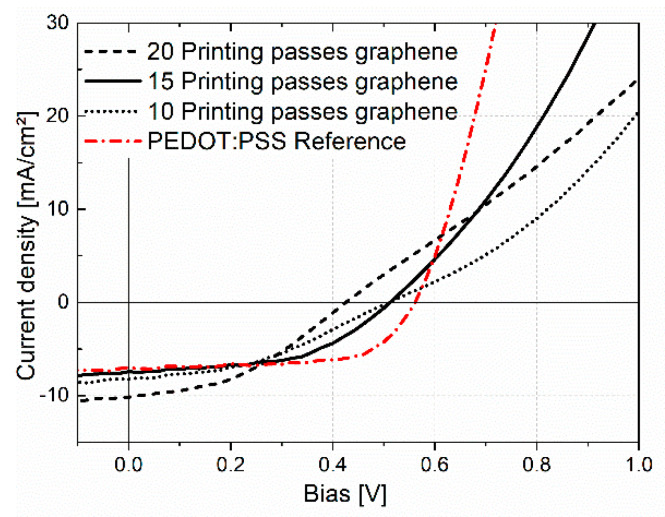
Typical *J*-*V* curves of the OPV cells with 10, 15 and 20 printing passes, respectively, of graphene and the reference PEDOT:PSS as a hole-transport layer.

**Figure 5 micromachines-14-01858-f005:**
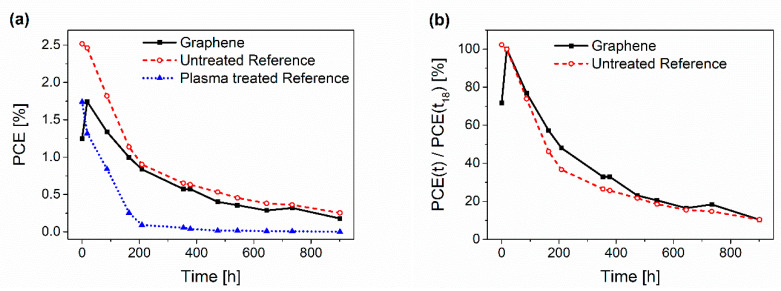
(**a**) Average degradation of the PCE of the solar cells with graphene as an HTL, the untreated reference cells with PEDOT:PSS and the plasma-treated reference cell with coated PEDOT:PSS. (**b**) Normalized degradation curves for non-encapsulated solar cells with graphene and PEDOT:PSS, respectively, stored in DH conditions.

**Figure 6 micromachines-14-01858-f006:**
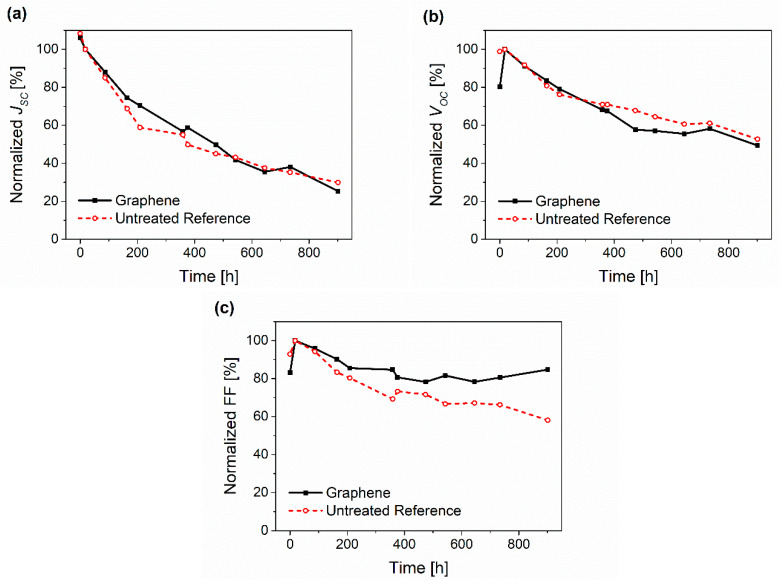
Comparative decay of the normalized (**a**) *J_SC_*, (**b**) *V_OC_* and (**c**) FF under DH storage.

**Figure 7 micromachines-14-01858-f007:**
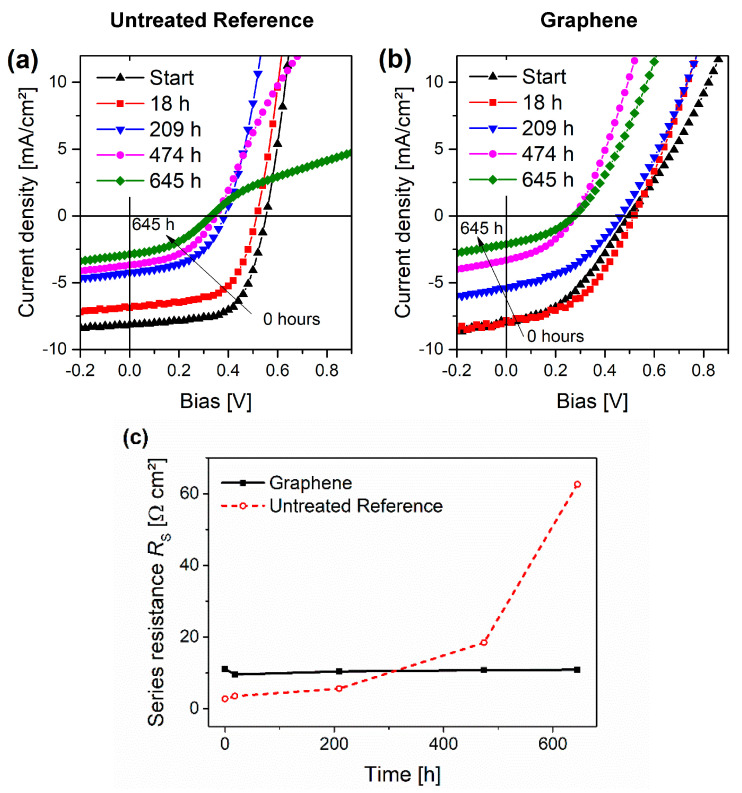
Typical *J*-*V* curve evolution during degradation from 0 h to 645 h in DH storage of (**a**) the untreated reference cells and (**b**) cells using printed graphene on the plasma-treated active layer as the HTL. (**c**) *R*_s_ was calculated and plotted for all available JV-curves at those time stamps.

**Table 1 micromachines-14-01858-t001:** Open circuit voltage (*V_OC_*), short circuit current (*J_SC_*), fill factor (FF) and maximum power conversion efficiency (PCE) of the devices depending on material and printing passes. The average values are calculated from 6–11 cells per category.

HTL	Printing Passes	*J_SC_* [mA/cm^2^]	*V_OC_* [V]	FF [%]	Average PCE Max [%]	PCE Max [%]
PEDOT:PSS	---	7.07 ± 0.22	0.54 ± 0.01	63 ± 4	2.41 ± 0.15	2.59
graphene	10	7.55 ± 0.47	0.50 ± 0.03	44 ± 4	1.65 ± 0.15	1.85
graphene	15	7.44 ± 0.11	0.51 ± 0.01	48 ± 3	1.82 ± 0.14	2.00
graphene	20	10.21 ± 0.71	0.41 ± 0.01	41 ± 1	1.72 ± 0.12	1.90

## Data Availability

Not Applicable.

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
