# Peer review of "Integration of Inkjet Printed Graphene as a Hole Transport Layer in Organic Solar Cells"

_micromachines, 2023, doi:10.3390/mi14101858_

Round 1

Reviewer 1 Report

The manuscript „Integration of Inkjet Printed Graphene as Hole Transport Layer in Organic Solar Cells“ by Julia Kastner, Flavia Tomarchio, Nicolas Decorde, Matthias Kehrer, Günter Hesser, and Anita Fuchsbauer describes the development of a graphene containing ink that can be used to inkjet-print hole transporting layer in organic solar cells. The results presented in this article could be important for an eventual industrialization of organic solar cells, particularly because it also tests the use of this graphene ink under ambient conditions and while employing so-called green solvents, which have garnered increasing attention lately as well. However, before publication may take place, the authors need to satisfactorily address the following points.

11.       In Fig. 1 the authors depict the conventional and inverted architecture of organic solar cells. In this case, it would be advisable to keep the images in a more generalized way, i.e. not describing what type of ETL or HTL is being used. Furthermore, usually inverted devices employed a layer sequence of ITO/ZnO/Active Layer/MoO3/Ag, not simply inverting the layer sequence of conventional devices. Furthermore, ZnO is not frequently used as ETL in conventional devices. Recently, organic compounds such as the PDINN, PDINO, or PDINO:PEIE have been used extensively as ETL (DOI: 10.1002/ente.202100281).

22.       On page 3, line 87, were the AFM measurements performed on a 20nm x 20nm area? This seems to be quite small.

33.       On page 6, lines 204-206, the authors mention the optical spacer effect. Here the authors should briefly describe this effect in a couple of sentences.

44.       In Fig. 7 (and other plots) the authors should consider to label their plots (i.e. Reference and Graphene…).

55.       Have the authors considered to measure the dark JV curves? This could be used to determine the differential resistance (similarly to obtaining the series resistance), and thus also the shunt resistance, which can yield further information on the device performance (see Fig. 2 of DOI: 10.1002/aenm.201901438).

66.       Have the authors considered to test their graphene ink on high performing, non-fullerene organic solar cells? The results with P3HT:PCBM are interesting, but can they also be transferred to the recently used systems based on, for instance, PM6 and Y6 (DOI: 10.1039/D1QM00060H)? In general, the authors should list in more details the latest developments in organic solar cells (i.e. non-fullerenes, ternary systems, switch to green solvents, etc.).

77.       Have the authors considered to perform light intensity dependent JV tests on their organic solar cells (Jsc vs P_light and Voc vs P_light)? This relatively simple method allows to determine qualitatively the dominant recombination dynamics of the tested devices, i.e. whether bimolecular, bulk-trap assisted or surface-trap assisted recombination are present and whether they dominate the loss processes (DOIs: 10.1016/j.progpolymsci.2013.08.008; 10.1016/j.orgel.2020.105905). It can also be used to estimate the role of the shunt resistance regarding the overall performance (this would then line up with point 5 above).

88.       There are some further points related to typos, spelling, and style that can be found in the commented manuscript file.

There are only a couple of minor typos and spelling mistakes.

Reviewer 2 Report

In this manuscript, the authors presented their work on inverted organic solar cells using inkjet-printed graphene HTLs with long-term stability.

While the initial PCE of graphene device is lower than the reference using PEDOT HTL, the graphene HTL shows promise in extending the lifetime of cells after long term. The main concerns for the work are (1) the limited stability improvement with mechanisms/physics not clearly explaiend (also the concerns of the appearance of s-shape in the initial JV in graphene cells), and (2) in the introduction, not enough justifications are made to distinguish graphene HTL from GO/rGO (e.g. as the later discussion shows, the work function of graphene compared to GO may prohibit its function as HTL). It would be beneficial for the authors to address these aspects.

Other minor issues: 

1. text from line 64-69 should be removed

2. in the caption of SI Fig1, is "vertical" supposed to be horizontal?

The writing of the manuscript should be carefully polished and revised by the authors. 

Reviewer 3 Report

This work shares a new method of printing graphite as a novel HTL to replace PEDOT:PSS. As a result, the P3HT:PCBM system based OSC devices exhibit a comparable PCE to that of PEDOT:PSS based counterparts. Furthermore, graphite HTL can break the bottleneck of Jsc value, though fails in PCE promotion, which implies huge potential of this material and method. Overall, the work can be published after taking care of some necessary issues:

[1] The abstract writing is not perfect. Device architecture is not suggested to be specialized here, while the explanation/mechanism of the graphite functional layer. Considerable revision is supposed to be done here.

[2] The introduction part missed mentioning the advance of OSC‘s PCE development. Currently, it has been more than 18%. This point should be taken care of in revision, as well as some related references:  Adv. Energy Mater. 2022, 12, 2201076.; Energy Environ. Sci. 2022, 15, 2479.; Sci. China Chem. 2021, 64, 581.;  Energy Environ. Sci. 2023, 16, 2316. Adv. Mater. 2023, 35, 2208986.; Energy Environ. Sci. 2022, 15, 4601.; Adv. Mater. 2022, 34, 2202089.; Adv. Mater. 2023, 35, 2301583.; Adv. Energy Mater. 2021, 11, 2003441. Note these are just some suggestions, more references with high PCEs shall be cited.

[3] Likewise, some HTL works on OSC shall be mentioned:  Adv. Energy Mater. 2021, 11, 2100492.; Adv. Mater. 2023, 35, 2212275. ACS Energy Lett. 2020, 5, 2935.; Advanced Functional Materials 2022, 32, 2205398.

[4] For the stability measurement, more details about testing condition shall be provided.

Apparently, the language presentation can be further improved by a proofreading.

Author Response

We thank for the review. For the responses, please see attachement.

Round 2

Reviewer 1 Report

The revised manuscript has been improved by the authors and all relevant corrections and additions have been performend to satisfactorily address the concerns raised by the various reviewers. Hence, the publication of the present, revised version of the manuscript should take place.